# Genetic ablation of interleukin-17A augments fibrosis in a mouse model of cholestatic liver injury

Takashi Kitagataya[1,2], Anuradha Krishnan[1], Kirsta E. Olson[1], Florencia Gutierrez[1], Michelle Baez-Faria[1], Maria Eugenia Guicciardi[1], Kevin D. Pavelko [3], Adiba I. Azad[1], Gregory J. Gores [1]*

1 Division of Gastroenterology and Hepatology, Rochester, Minnesota, United States of America,
2 Department of Gastroenterology and Hepatology, Graduate School of Medicine, Hokkaido University, Sapporo, Japan, 3 Department of Immunology, Mayo Clinic, Rochester, Minnesota, United States of America

* gores.gregory@mayo.edu

## Abstract

### Aim

The underlying mechanisms contributing to cholestatic liver injury remain unclear. The pro-inflammatory leukocyte-restricted cytokine interleukin-17A (IL-17A) has been implicated in human cholestatic liver injury. However, mechanistic insights are lacking and require further exploration in preclinical models. Herein, we examined the effect of IL-17A genetic ablation in a mouse model of cholestatic liver injury.

### Method

Age and gender-matched littermate wild type (WT) and $Il$-$17a^{-/-}$ C57BL/6 mice were fed an intermittent 0.1% 3,5-diethoxycarbonyl-1,4-dihydrocollidine (DDC) diet for 21 days to induce cholestatic liver injury or a control diet.

### Results

As compared to WT littermates, $Il$-$17a^{-/-}$ mice displayed more abundant desmin-positive myofibroblasts and increased fibrosis. NanoString analysis of intrahepatic leukocyte populations using a fibrosis-related gene panel identified upregulation of $Tnfsf14$ (encoding the protein LIGHT) in the DDC-fed $Il$-$17a^{-/-}$ mice. Although mass cytometry identified an increase in myeloid cells in both genotypes of the DDC-fed mice, we could not identify LIGHT expression in this cell lineage. Instead, the upregulation of LIGHT expression was largely restricted to a CD4+ T cell population as assessed by flow cytometry. Enhanced LIGHT expression was observed in a Th1+ CD4+ T cell population. LIGHT activated primary human hepatic stellate cells $in$ $vitro,$ suggesting that LIGHT stimulation of hepatic fibrogenesis may be direct.

**Data availability statement:** Data supporting the findings of this study are available within the article and in the supplementary information.

**Funding:** This work was supported by NIDDK-funded grant DK124182 (GJG), the NIDDK-funded Optical Microscopy Core of the Mayo Clinic Center for Cell Signaling in Gastroenterology (P30DK084567), and the Mayo Clinic, Rochester. The funders had no role in study design, data collection and analysis, decision to publish, or preparation of the manuscript.

**Competing interests:** The authors have declared that no competing interests exist.

**Abbreviations:** ALP, Alkaline phosphatase; ALT, Alanine transferase; BA, Bile acids; CK7/19, Cytokeratin 7/19; CyTOF, Cytometry Time of Flight (High dimensional mass cytometry); DDC-3,5, Diethoxycarbonyl-1,4-dihydrocollidine; DR, Ductular reaction; IL-17, Interleukin-17; IHL, Intrahepatic leukocytes; PSC, Primary Sclerosing Cholangitis; qRT-PCR, Quantitative real-time PCR; TBIL, Total bilirubin; t-SNE-t, distributed Stochastic Neighbor Embedding.

## Conclusion

Taken together, these data suggest that IL-17A restrains expression of the profibro-genic cytokine, LIGHT, by Th1-polarized CD4$^+$ T cells, and implicate a role for LIGHT in cholestatic fibrogenesis in DDC-fed mice; a finding which requires validation in additional models.

## Introduction

Interleukin-17A (IL-17A) has not only emerged as a central mediator of inflammation and tissue damage in various immune-mediated and inflammatory diseases but also as a successful therapeutic target for several of these conditions [1–3]. Recently, there has been a growing focus on the role of IL-17A in a variety of liver diseases, with studies implicating its involvement in the pathogenesis of liver injury and fibrosis, including cholestatic liver injury [4]. Primary sclerosing cholangitis (PSC) is an example of an important immune-mediated, human cholestatic liver disease with limited therapeutic options [5,6], in which IL-17A has been implicated in its pathogenesis [7–10].

IL-17A, the most prominent member of the IL-17 family (IL-17A-IL-17F), shares structural similarities and overlapping biological effects with IL-17F. However, IL-17A has been demonstrated to play a dominant role in autoimmunity, thus making it the primary target for IL-17-directed therapy [11,12]. IL-17A is primarily produced by CD4$^+$ T helper (Th17) cells, although it is also produced by other CD4$^+$ T cell subsets and non-CD4$^+$ T cells, such as γδ T cells, CD8$^+$ T cells, natural killer T (NK T) cells, innate lymphoid cells, and neutrophils [13]. After binding to its corresponding receptor family (IL-17RA and IL-17RC), IL-17A exerts its pro-inflammatory effects through the activation of downstream signaling pathways, including nuclear factor kappa-light-chain-enhancer of activated B cells (NF-κB) and mitogen-activated protein kinase (MAPK) pathways [11]. Through these signaling cascades, IL-17A drives the production of pro-inflammatory cytokines, chemokines, anti-microbial peptides (AMPs), and matrix metalloproteinases (MMPs), which play a critical role in host defense and tissue repair [14]. Nevertheless, in the context of cholestatic diseases, these identical molecules may create an environment that enables persistent inflammation, subsequent tissue damage, and fibrosis. For example, stimulation with IL-17A upregulates the expression of IL-6, IL-1β, IL-23, CCL20, and CCL2 in human cholangiocytes, subsequently attracting more Th17 cells to the liver parenchyma [15]. This stimulation also prompts Kupffer cells, the resident macrophages in the liver, and hepatic stellate cells to respond by increasing the expression of pro-inflammatory cytokines (IL-6, IL-1β, TNFα), profibrotic cytokines (TGFβ), and activation markers [16,17].

While all these studies indicate a detrimental effect of IL-17A, it is important to take into consideration that there is an association between PSC and inflammatory bowel disease (IBD), which is present in approximately 70% of PSC patients [5]. IL-17A-directed therapy has proven to be unsuccessful and even detrimental in IBD due to the crucial role of IL-17A in maintaining intestinal epithelial permeability and

mucosal integrity [18–20]. Furthermore, a recent study described that by upregulating programmed cell death ligand 1 (PD-L1) on cholangiocytes, IL-17 played an essential role in restricting cholestatic liver injury by protecting against CD8+ T cell-mediated, inflammatory bile duct injury [21]. These data indicate that the functional role of IL-17A varies depending upon the disease context. Therefore, in this study, we aimed to assess the impact of IL-17A genetic deletion on the liver of DDC-fed mice, a model that phenocopies many of the features of chronic cholestatic liver injury. Taken together, our data suggests that IL-17A modulates the expression of the profibrogenic cytokine LIGHT by Th1-polarized CD4+ T cells in a mouse model of cholestasis, thereby restricting liver fibrosis. These data provide insights into the role of CD4+ T cell subsets in cholestatic liver fibrosis and have implications for the treatment of human cholestatic liver disease by targeting LIGHT or specific CD4+ T cell subsets.

## Materials and methods

### Animal studies

All animal experiments were ethically reviewed and performed according to a protocol (Number A00004446-19-R22) approved by the Institutional Animal Care and Use Committee of the Mayo Clinic, Rochester, MN. All procedures were compliant with the US National Research Council's "Guide for the Care and Use of Laboratory Animals" and with the US Public Health Service's Policy on "Humane Care and Use of Laboratory Animals". C57BL/6 wild-type mice, *Il-17a*tm1.1(icre)Stck mice (*Il-17a*-/-), and *Il-17af*-/- on C57BL/6 background were obtained from Jackson Laboratory (Bar Harbor, ME). *Il-17a*-/- and *Il-17af*-/- littermate wild-type mice were generated by backcrossing heterozygous *Il-17a*+/- or *Il-17af*+/- mice. Genomic deletion of IL-17A was confirmed by genotyping using DNA isolated from tail clips (S1A Fig). Both male and female mice were used and were randomly assigned to either the cholestatic injury or control treatment groups (n = 10 per group). All genotypes were confirmed prior to use. Cholestatic liver injury was induced in 8–10 weeks old, gender-matched experimental mice by feeding a diet containing 0.1% DDC (104242 GI, Dyet Inc., Bethlehem, PA, USA). Control groups were fed the control diet (104242, Dyet Inc., Bethlehem, PA, USA). The diets were provided intermittently for 5 days each (3 cycles), alternating with 3 days of standard rodent chow (2 cycles of Rodent Diet 5053, Pico Lab diets) for a total of twenty-one days (Fig 1A). Mice were euthanized by carbon-dioxide inhalation on day 22. Body weight was noted. Blood for serum analysis was drawn via cardiac puncture. The liver was flushed with PBS via the heart, isolated, and weighed. Samples of each lobe were immediately preserved in 10% neutral buffered formalin (245–684, Fisher HealthCare) or snap-frozen for other downstream applications.

### Serum biochemistry

Blood samples were centrifuged at 10,000 rpm for 20 minutes to separate the serum. Serum samples were analyzed for alanine transferase (ALT), alkaline phosphatase (ALP), bile acids (BA), total bilirubin (TBIL), and cholesterol using the mammalian liver profile rotor (500–7128; Abaxis) on the Vetscan VS2 (Abaxis, Union City, CA).

### Histology

Samples of the left, right, and caudate lobes were fixed in 10% neutral buffered formalin. After 48 hours, tissue samples were processed and paraffin embedded using standard methods at the Biomaterials and Histomorphometry Core Facility, Mayo Clinic, Rochester, MN. Tissue sections (5 µm) were stained with hematoxylin-eosin (H&E). Fibrosis was assessed by picrosirius red staining (365548, Sigma) as previously described [22] and quantified directly under plane-polarized light using an inverted microscope (Nikon Eclipse TE300, Nikon, Japan) at a magnification of 10X. The Nikon AR software v2.3 was used to quantify collagen area as a percentage of field of view.

### Fluorescence Immunostaining

Tissue sections were deparaffinized with xylene and hydrated over ethanol gradient. Antigen retrieval was performed by boiling in 10 mM sodium citrate buffer (pH 6.0) for 20 minutes. The tissue sections were blocked with 5% bovine serum

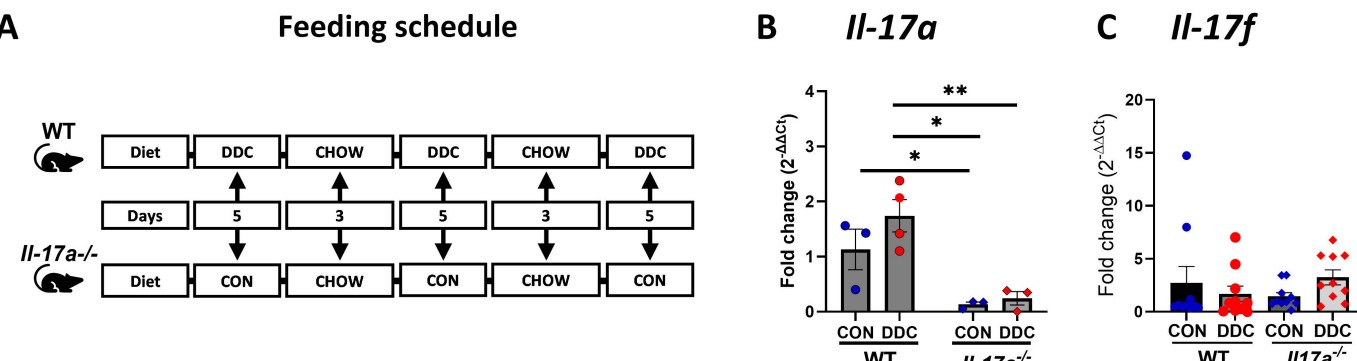

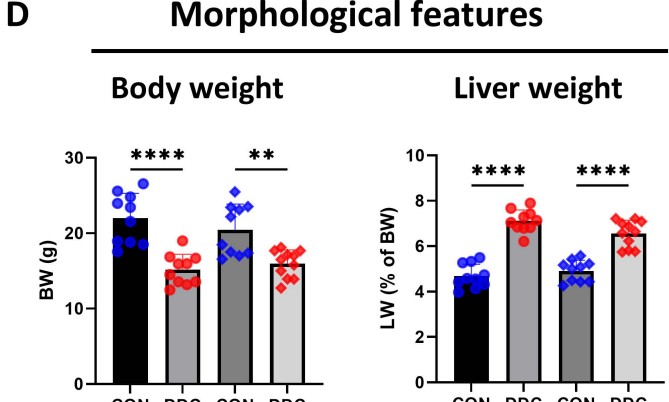

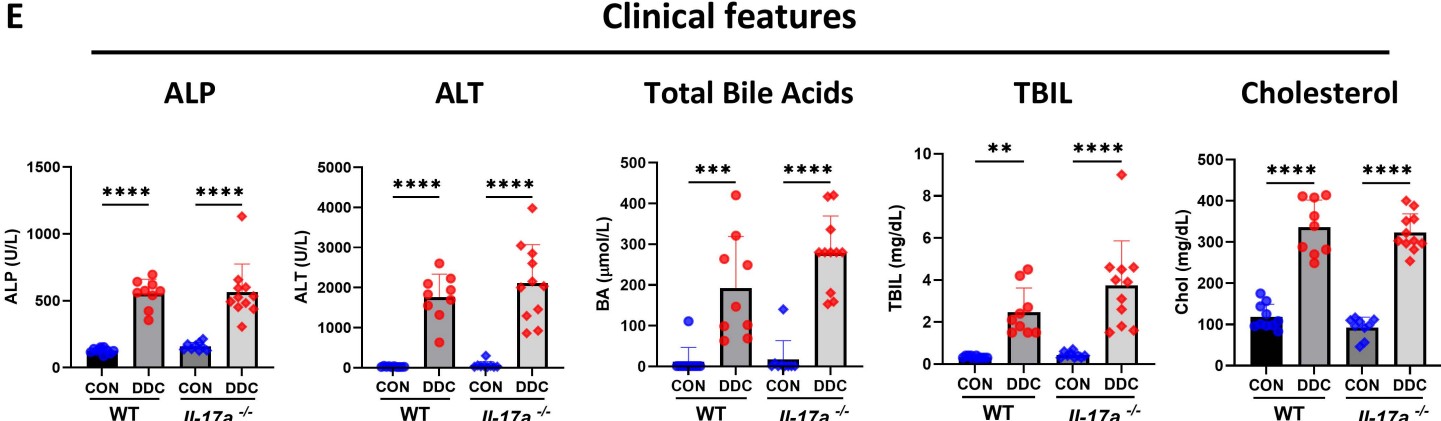

**Fig 1. DDC diet-induced liver injury is comparable in WT and *Il-17a*-/- mice. A.** An illustration of the intermittent feeding schedule alternating between DDC/Control and Chow diets for 21 days (n = 10 per group). **B.** Gene expression of *Il-17a* is diminished in the IHL of the *Il-17a$^{-/-}$* mice (N = 3-4 per group). **C.** Gene expression of *Il-17f* is similar between WT and *Il-17a$^{-/-}$* mice (N = 10 per group). **D.** Gross morphological features viz., body weight and liver weight display similar trends between in cholestatic WT and *Il-17a$^{-/-}$* mice (N = 10-11 per group). **E.** Clinical indicators of liver injury indicate significant difference between mice fed control and DDC diets within each genotype but there was no difference between DDC diet-fed WT and *Il-17a$^{-/-}$* mice (N = 10-11 per group). * - $p < 0.05$, ** - $p < 0.01$, *** - $p < 0.005$, **** - $p < 0.001$.

albumin (A9647, Sigma-Aldrich) or Rodent Block M (RBM961H, Biocare Medical) and incubated overnight at 4°C with the primary antibodies (S1 Table). Fluorophore-tagged species-specific secondary antibodies were used to detect bound primary antibodies at a dilution of 1:200. For immunostaining of fibrillar collagen, Image-iT™ FX signal enhancer (I36933, ThermoFisher Scientific) was applied to tissue sections per manufacturer's instructions. Subsequently, sections were stained with 4 ′, 6-Diamidino-2-phenylindole (DAPI, 1 µg/mL) solution to visualize nuclei. For quantification, a minimum of 10 fields of view (FOV) images were acquired per section on an EVOS M5000 (ThermoFisher Scientific, Waltham, MA) or a confocal microscope (LSM 980, Zeiss, Germany). Digital image analysis was performed using the ImageJ software (NIH, MD). Tile scan images for CK19 were acquired at 20x magnification on a Zeiss Axio Observer microscope (Carl Zeiss, Thornwood, NY) equipped with a computer-controlled stage, a Colibri 7 LED light source, a Zeiss Axiocam 702 digital camera all controlled with Zen software (Carl Zeiss, Thornwood, NY). Multiple images were acquired and stitched together to make a single large high-resolution image using the tiling feature of the software. As tissue size differed between sections, a scale bar representing 1000 µm was included in each image for comparison. For serial immunostaining involving more than two antigens on the same tissue section, antibody stripping was performed with the 2-mercaptoethanol/SDS (2-ME-SDS) method after image acquisition of the previous staining [23]. Images were acquired capturing x and y coordinates. The images acquired on every channel were merged and aligned in ZEN software (blue edition ver.3.5, Zeiss, Germany) using DAPI staining and the x and y coordinates as references. Staining for all antibodies was optimized to ensure minimal background and verified for specificity using secondary antibodies alone.

### Isolation of intrahepatic leukocytes (IHL)

IHL were isolated from littermate WT mice and *Il-17a^-/-* mice by Percoll density gradient centrifugation as outlined previously [22].

### Quantitative real-time PCR (qRT-PCR)

Total RNA was isolated from IHL or HSC using RNeasy Plus Mini Kit (74104, Qiagen). Total RNA from liver fragments was isolated with TRIzol using standard techniques. One µg of total RNA was used to transcribe complementary DNA (cDNA) using the iScript cDNA Synthesis Kit (1708891, Bio-Rad). Quantitative real-time PCR was performed with a QuantStudio6 Flex Real-Time PCR System (Applied Biosystems, Waltham, MA) using Power Up SYBR Green PCR Master Mix (4367659, Applied Biosystems). Primers are listed in S2 Table. Gene expression was calculated using the $2^{-\Delta\Delta Ct}$ method and expressed as fold change.

### Multiplex Gene Expression Assay

Total RNA was extracted from freshly isolated IHL from WT and *Il-17a^-/-* mice (N = 3 per group), and 100 ng of total RNA was used to determine the expression of a panel of fibrosis-related genes using the nCounter Mouse Fibrosis V2 Panel (NanoString Technologies, Seattle, USA) according to the manufacturer's instructions, employing an nCounter MAX analysis system (NanoString Technologies, Seattle, WA). Data were acquired as RNA copy numbers.

### Time-of-flight Mass cytometry (CyTOF)

Freshly isolated IHL from WT and *Il-17a^-/-* mice (N = 3 per group) were stained with a cocktail comprising thirty-seven antibodies as described previously [22]. The list of antibodies is detailed in S3 Table.

### Flow cytometry

WT and *Il-17a^-/-* mice (N = 3 per group) were fed with the DDC diet for five consecutive days. IHL were isolated and counted on a cell counter (Cellometer®, Nexcelom Bioscience, Lawrence, MA), prepared as a single cell suspension,

---

and resuspended in 1% FBS in PBS. Cells were stained with fluorochrome-conjugated antibodies in 1% FBS in PBS (S1 Table). Intracellular staining for transcription factors (T-BET for Th1, GATA3 for Th2, RORγ(t) for Th17, and FOXP3 for Treg) and LIGHT was performed using Foxp3 Transcription Factor Staining Buffer Set (eBioscience, ThermoFisher Scientific) following the manufacturer's instructions. Flow cytometry acquisition and analysis were performed on a MACSQUANT Analyzer 10 (Miltenyi Biotech) using appropriate fluorescence minus one (FMO) controls. Data were analyzed on FlowJo software v10.0 (BD Life Sciences).

In separate experiments, splenocytes were isolated from WT (N = 3) and *Il-17a*<sup>-/-</sup> (N = 4) mice and stimulated either with a combination of phorbol 12-myristate 13-acetate (PMA, 50 ng/ml) and ionomycin (1 µg/ml) or with vehicle alone for a period of 4 hours in the presence of brefeldin (5 ng/ml). Splenocytes were stained with fluorochrome conjugated antibodies for cell surface markers CD3 and CD4. Intracellular staining for nuclear transcription factor T-BET, IFNγ and for LIGHT was performed as mentioned above. Flow cytometry data was acquired on a FACS Canto X SORP (BD Bioscience), at the Microscopy and Cell Analysis Core of the Mayo Clinic, Rochester, MN. Data were analyzed on the FlowJo software v 10.0 (BD Life Sciences).

### Primary human hepatic stellate cell culture

Primary human hepatic stellate cells (5300, SciCell Research Laboratories, Carlsbad, CA, USA) were cultured in Dulbecco's modified Eagle medium supplemented with 10% fetal bovine serum, and normocin at 100 µg/mL. Cells at passage 2 were seeded at a density of 77,000 cells in 30 µl of Matrigel (356231, Corning, Glendale, AZ, USA) in a 48 well tissue culture plate with 230 µl of medium. Cells were rested for 4 hours and stimulated with medium containing vehicle or recombinant human LIGHT (664-LI, R&D systems, Minneapolis, MN) for 72 hours refreshing the medium at 48 hours. Total RNA was isolated using the Direct-zol RNA microprep kit (R2060, Zymo Research, Irvine, CA). qRT-PCR was performed as outlined above on a QuantStudio6 Flex Real-Time PCR System (Applied Biosystems, Waltham, MA) using Power Up SYBR Green PCR Master Mix (4367659, Applied Biosystems). Primers are listed in supplemental S2 Table.

### Statistical analysis

Data are expressed as mean ± SEM representing replicates within an experiment. If not mentioned otherwise, the ordinary one-way ANOVA test and Tukey's post-test determined statistical significance between multiple groups. The Mann-Whitney test using GraphPad Prism (version 9.3.1, GraphPad Software, San Diego, CA) defined the statistical difference between the two groups.

## Results

### DDC diet-induced cholestatic injury and ductular reaction are comparable in WT and *Il-17a-/-* mice

Mice were fed a control or DDC diet for 21 days as described in materials and methods (Fig 1A). Lack of *Il-17a* expression in *Il-17a*<sup>-/-</sup> IHL was verified by qRT-PCR (Fig 1B) and by immunostaining for IL17-A (S1B Fig). The absence of compensation by *Il-17f* was also excluded by qRT-PCR for its mRNA, albeit this was not confirmed at the protein level (Fig. 1C). The WT and *Il-17a*<sup>-/-</sup> mice on the DDC diet had lower body weight and, consequently, higher liver-to-body weight ratios that were comparable in both genotypes (Fig 1D). Additionally, serum levels of ALT, ALP, total bile acids, total bilirubin, and cholesterol were similar in the WT and *Il-17a*<sup>-/-</sup> DDC-fed mice (Fig 1E).

Periportal inflammation was evaluated by H&E staining and appeared similar between the DDC-fed WT and *Il-17a*<sup>-/-</sup> mice (Fig 2A). As the expansion of ductular reactive (DR) cells is one of the histopathological characteristics of cholangiopathies [24], DR cell expansion was examined by immunostaining for the markers, cytokeratin 19 (CK19) and cytokeratin 7 (CK7). CK19-positive and CK7-positive areas were increased in both WT and *Il-17a*<sup>-/-</sup> mice on the DDC diet (Figs 2B and 2C). Collectively, these data indicate that the genetic ablation of *Il-17a* did not alter the extent of ductular reaction in DDC diet-induced cholestasis in the WT and *Il-17a*<sup>-/-</sup> mice.

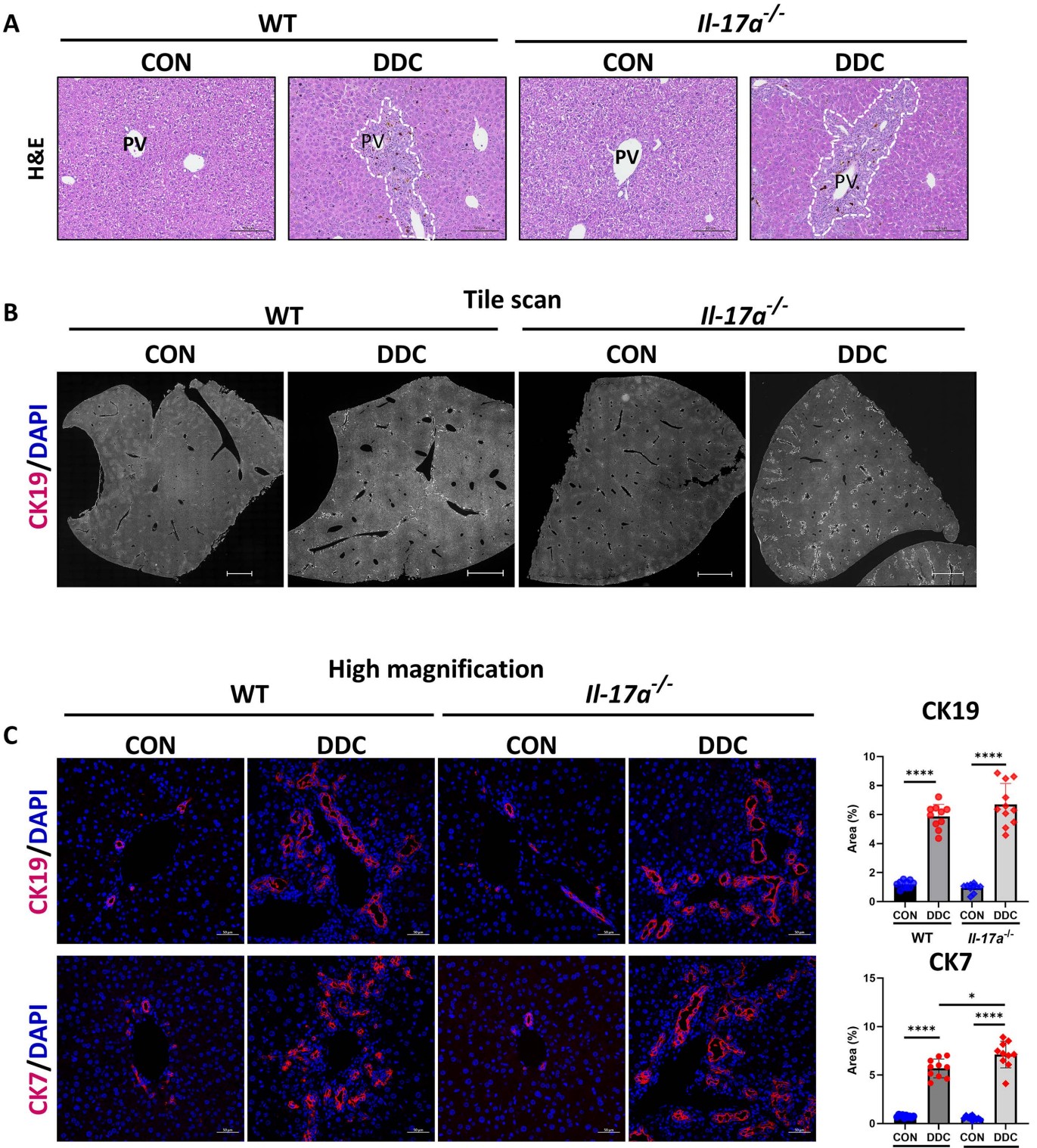

**Fig 2. Histological evidence of DDC diet-induced injury and ductular reaction in WT and *Il-17a-/-* mice. A.** Representative images of FPPE liver tissue sections stained with hematoxylin and eosin. Periductular inflammation in the liver of cholestatic WT and *Il-17a⁻/⁻* mice is outlined by a dashed white line. PV – Portal vein. Scale bar = 50 µm. **B.** Representative tile scan (a composite image of multiple small tiles covering the entire tissue section)

images of FPPE liver tissue sections immunostained for the cholangiocyte marker CK19. Scale bar = 1000 μm. **C.** Representative images of FPPE liver tissue sections immunostained for the cholangiocyte marker CK19 (Top panel) and CK7 (bottom panel). Scale bar = 50 μm. The panels on the right provide digital image quantification of CK19 and CK7 (N = 10-11 per group). * - p < 0.05, ** - p < 0.01, *** - p < 0.005, **** - p < 0.001.

## Hepatic myofibroblasts are increased, and hepatic fibrosis is enhanced in the cholestatic *Il-17a-/-* mice

To determine the effect of *Il-17a* deletion on hepatic fibrosis, we first stained liver tissue sections for desmin to detect the presence of myofibroblasts (Fig 3A). There was increased abundance of total myofibroblasts in the DDC-fed mice *Il-17a^{-/-}* mice compared to their WT counterparts. As deposition of fibrillar collagen is the hallmark of hepatic fibrosis [25,26], we analyzed Sirius red-stained liver tissue sections under plane-polarized light (Fig 3B) and additionally quantified COL1α1 immunostained areas (Fig 3C). We have previously noted that there is no compensatory increase in the *Il-17f* in the *Il-17a^{-/-}* mice (Fig 1C). Hence, we examined the expression of a panel of pro-fibrotic genes in the whole liver lysates of *Il-17af^{-/-}* mice. The genes *Tgfb, Timp1, Fn1, Acta2, Col1a1,* and *Col4a* were significantly upregulated in the cholestatic *Il-17af^{-/-}* mice as compared to their WT cholestatic littermates (Fig 3D). Taken together, these data indicate that the deletion of *Il-17a* aggravates portal fibrosis in cholestatic liver injury.

## DDC diet-induced cholestasis drives a transformation in the immune cell landscape of WT and *Il-17a-/-* mice

A visualization of the periportal tract of cholestatic WT mouse liver by serial immunostaining indicated extensive deposits of collagen and the presence of multiple cell types: T cells, macrophages, neutrophils, and fibroblasts proximal to the cholangiocytes in the periductal area (S2 Fig). To determine how immune cells contributed to the enhanced fibrosis observed in the mutant mice, we used an unbiased high-dimensional platform, viz., time-of-flight mass cytometry (CyTOF), to evaluate the immune cell landscape in WT and *Il-17a^{-/-}* mice raised on either the control or DDC diets. IHL were isolated and labeled with an antibody cocktail comprising thirty-seven markers (S3 Table) that could differentiate lymphoid and myeloid cell populations. Non-linear dimensionality reduction plots generated by t-SNE mapping indicated that the CD45^+ IHL aligned and sorted into twenty-eight distinct clusters (Fig 4A) with unique cluster identity (S3 Fig). These comprised multiple clusters of T cells (1, 5, 7, 10, 19, 20, 21, 28), B cells (2, 6, 8, 14, 17, 22), Kupffer cells (3), macrophages (23, 24, 26), neutrophils (13, 25, 27), dendritic cells (4, 9, 11) and other immune cells that were CD45^+ but bore no distinct cell type marker (12, 15, 16) (Fig 4A, S3 Fig). Hierarchical clustering analysis, as illustrated by Rphenographs (Fig 4B), demonstrated that the non-cholestatic and cholestatic mice segregated into two distinct groups (Fig 4B, left panel = samples) and highlighted relationships between specific cluster types (Fig 4B, right panel = markers). Importantly, individual t-SNE plots of WT and *Il-17a^{-/-}* mice (Fig 4C) revealed that cholestasis drove alterations in the immune cell composition and their relative proportions (S4 Fig). Under basal conditions, lymphocytes (B and T cells) predominated, together accounting for greater than 60% of the immune cells. However, with cholestasis there was an influx of myeloid cells that altered the immune landscape with neutrophils, macrophages and dendritic cells predominating. Remarkably, although many individual clusters were abundant in the cholestatic mice, no cluster predominated in either the WT or the *Il17a-/-* mice on the DDC diet (S4 Fig). However, when summed up along broad immune cell categories, total B cells, which accounted for about 49% of the IHL population in non-cholestatic livers, decreased to about 3% in cholestatic livers of WT and *Il-17a^{-/-}* mice (Fig 4D). On the other hand, while DDC diet-induced cholestasis resulted in a reduction of total T cells in the WT mice, the proportion of total T cells in the *Il-17a^{-/-}* mice remained unchanged (Fig 4D). Resident Kupffer cells decreased to <1% in cholestatic livers of both WT and *Il-17a^{-/-}* mice. By contrast, infiltrating macrophages and dendritic cells were equally increased in the DDC diet-induced cholestatic liver of both genotypes (Fig 4D). The myeloid-derived cells bore the markers of infiltrating immune cells, viz., positivity for Ly6C, CX3CR1, and high levels of CD11b (S3 Fig). The increased presence of macrophages and neutrophils was also confirmed by immunostaining for the pan-macrophage IBA1 (Fig 4E, upper panel) and the neutrophil marker, MPO (Fig 4E, lower panel). Taken together, these data suggest that cholestasis

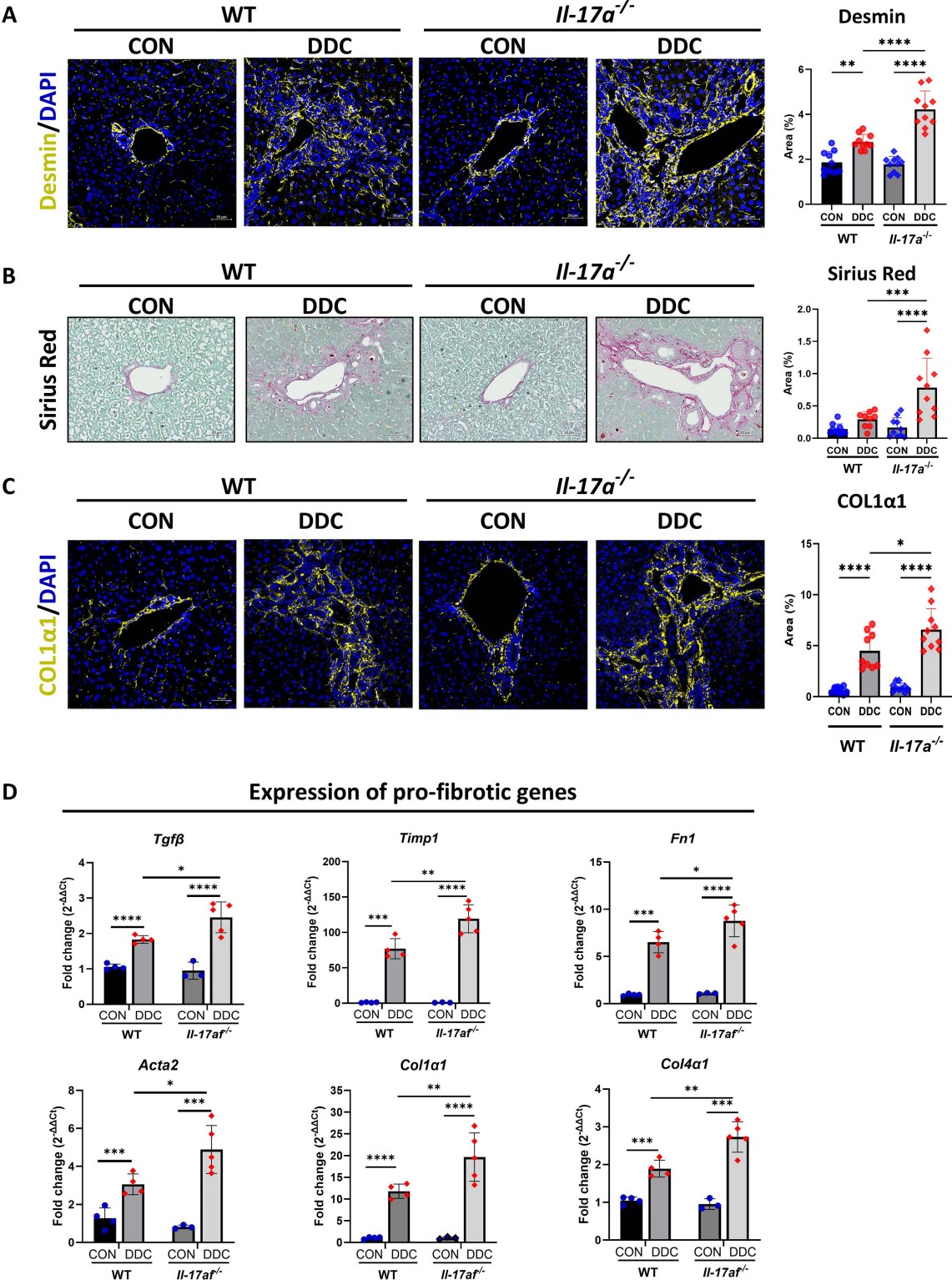

**Fig 3. Hepatic myofibroblasts and hepatic fibrosis are increased in the *Il-17a*-/- mice with DDC diet-induced obstructive cholestasis. A.** Representative images of FPPE liver tissue sections immunostained for the fibroblast marker, desmin (N = 10 per group), Scale bar = 50 μm. **B.** Representative images of FPPE liver tissue sections stained with Sirius red for detecting fibrosis (N = 10 per group). **C.** Representative images of FPPE liver tissue

sections immunostained for the fibrillar collagen marker COL1α1 (N = 5 per group). Scale bar = 50 μm. Digital image quantification for desmin, sirius red, and COL1α1 are provided in the corresponding panels on the right. **D.** Quantitative real-time PCR data for expression of multiple genes related to fibrosis in the *Il-17af-/-* mice (N = 3-5 per group). * - p < 0.05, ** - p < 0.01, *** - p < 0.005, **** - p < 0.001.

drives a reduction in the resident Kupffer cells and B cells and increases the proportion of infiltrating myeloid cells. Overall, there appeared to be a shift from a lympho-centric immune landscape observed in the normal hepatic environment to a myeloid-centric phenotype in the cholestatic liver that was comparable between the WT and the *Il-17a-/-* mice. Importantly, although total T cell population declined in the DDC-fed WT mice, their numbers remained unchanged in the *Il-17a-/-* mice.

### Immune cells from cholestatic mice *Il-17a-/-* express a profibrogenic gene signature that includes LIGHT (*Tnfsf14*)

The most apparent effect of the deletion of *Il-17a* on DDC diet-induced cholestatic liver injury was enhanced fibrosis observed in the mutant mice. To better understand whether the altered immune cell environment in DDC diet-induced injury promoted a profibrogenic response, we performed a curated, commercially available multiplex gene expression assay (NanoString nCounter) comprising over 760 genes related to multiple pathways associated with fibrosis, using total RNA isolated from the IHL. A heatmap of the differentially expressed genes in mice with and without cholestasis highlights the differential expression of genes between the WT and the *Il-17a-/-* mice (Fig 5A). Volcano plots of the differentially expressed genes between mice reared on control or DDC diets of WT (Fig 5B) and *Il-17a-/-* mice (Fig 5C) showed upregulation of similar genes involved in extracellular matrix synthesis and breakdown, including *Mmp12, Mmp8, Mmp7, Mmp13*, and *Fn1* when compared to their control diet-fed mice. Likewise, the cytokine, *Cxcl2,* and the DAMP, *S100a4,* were upregulated in WT and *Il-17a-/-* cholestatic mice. A volcano plot constructed to identify genes that were differentially regulated specifically in cholestatic WT and *Il-17a-/-* mice (Fig 5D) indicated that there were no differences in the expression levels of major pro-inflammatory cytokines such as *Ifng, Il1b* and *Tnf* between both groups. Notably, only seven genes, viz., *Il18r1, Trat1, Tnfsf14, Thbs1, Ccr4, Il2ra* and *Icos* were upregulated in the cholestatic *Il-17a-/-* mice (Fig 5E). The gene *Tnfsf14*, encoding for the protein LIGHT, is known to promote fibrosis in the liver, and indeed, the increased gene expression of *Tnfsf14* in the IHLs was confirmed by qRT-PCR (Fig 5F).

Next, we turned to flow cytometry to determine if LIGHT was expressed in a specific immune cell compartment. The gating strategy adopted for analysis is illustrated in S5A Fig. Total IHL was elevated in all cholestatic mice (Fig 6A), with greater numbers per gram liver weight for the *Il-17a-/-* mice (Fig 6B). CD45+ immune cells were detected equally in all mice reared on either the control or the DDC diet (Fig 6C). Among the immune cells, LIGHT positivity was observed only in the CD4+T and NK T cells, accounting for about 8% and 1% of the total CD45+ immune cells, respectively (Fig 6D, 6E). LIGHT was absent or minimally expressed by NK cells, CD8α+ T cells, neutrophils, dendritic cells, and macrophages (S5B-F Fig). As the LIGHT+ NK T cell population was low and did not alter with cholestasis (Fig 6E), we focused on the CD4+ T cells and their subsets. In keeping with the mass cytometry data, total CD4+ T cells were decreased in all cholestatic mice (Fig 6D, left panel). However, the abundance of LIGHT+ CD4 T cells trended toward increased abundance in cholestatic WT and *Il-17a-/-* mice (Fig 6D, right panel), suggesting that this subset of T cells may have contributed to the increased fibrosis associated with cholestasis. Further analysis indicated that although LIGHT was expressed by all subsets of CD4+ T cells, viz., Th1, Th2, Th17, and the T-regs (Fig 6F), LIGHT positivity significantly increased from about 50% to 100% only in the polarized Th1 subset in the cholestatic mice.

To more fully characterize the observation between IL-17A regulation of LIGHT expression in Th1 CD4+ T cells, WT mouse splenocytes were isolated and stimulated with PMA/ionomycin combination for maximal activation to induce IFNγ expression, a cardinal feature of Th1 polarized T cells. The PMA/ionomycin treated CD4 T cell population displayed an increased abundance of IFNγ+ CD4+ T cells with a corresponding twelve-fold increase in LIGHT+ cells in the IFNγ+ subpopulation as compared to no increase in the IFNγ- subpopulation (Fig 6G–I). Further, under control conditions, there was

an increase in the IFNγ⁺ CD4⁺ T cells in the *Il-17a⁻/⁻* mice as compared to the WT mice (S6B Fig). Accordingly, there was a significant increase in the proportion of LIGHT⁺ T cells in this subpopulation in the *Il-17a⁻/⁻* mice as compared to the WT mice (S6B Fig). Taken together, the data imply a relationship between LIGHT expression in Th1 CD4⁺ T cells, which is negatively regulated by IL-17A.

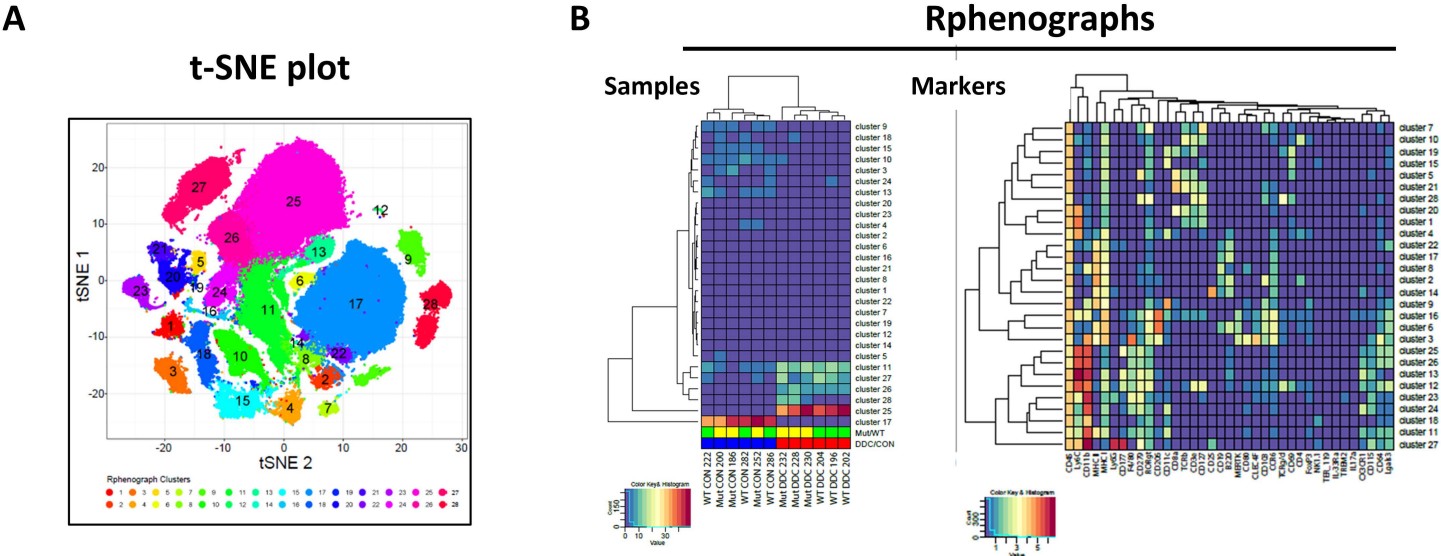

1. T cell, 2. B cell, 3. Kupffer cell, 4. Dendritic cell, 5. CD8+ T cell, 6. B cell, 7. T-reg, 8. B cell, 9. Dendritic cell, 10. CD4+ T cell, 11. Dendritic cell, 12. Unknown, 13. Neutrophil, 14. B cell, 15. Unknown, 16. Unknown, 17. B cell, 18. Monocyte, 19. Cd8+ T cell, 20. CD8+ T cell, 21. Cd8+ T cell, 22. B cell, 23. Macrophage, 24. Macrophage, 25. Neutrophil, 26. Macrophage, 27. Neutrophil, 28. γδ T cell

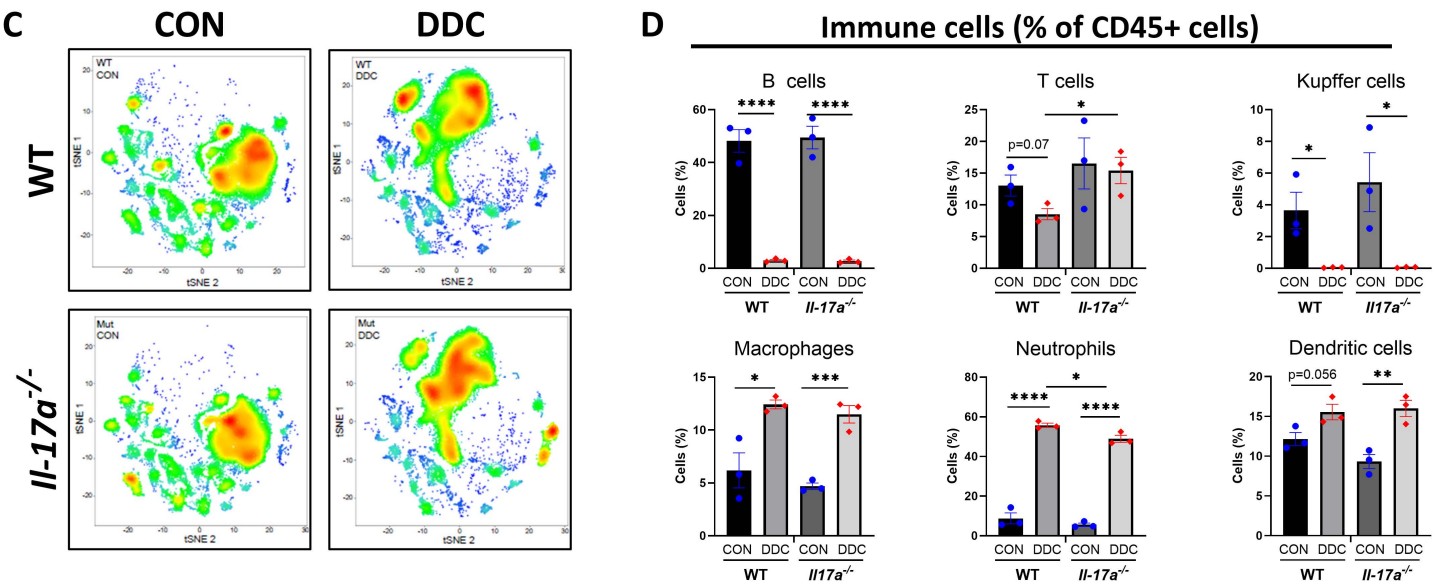

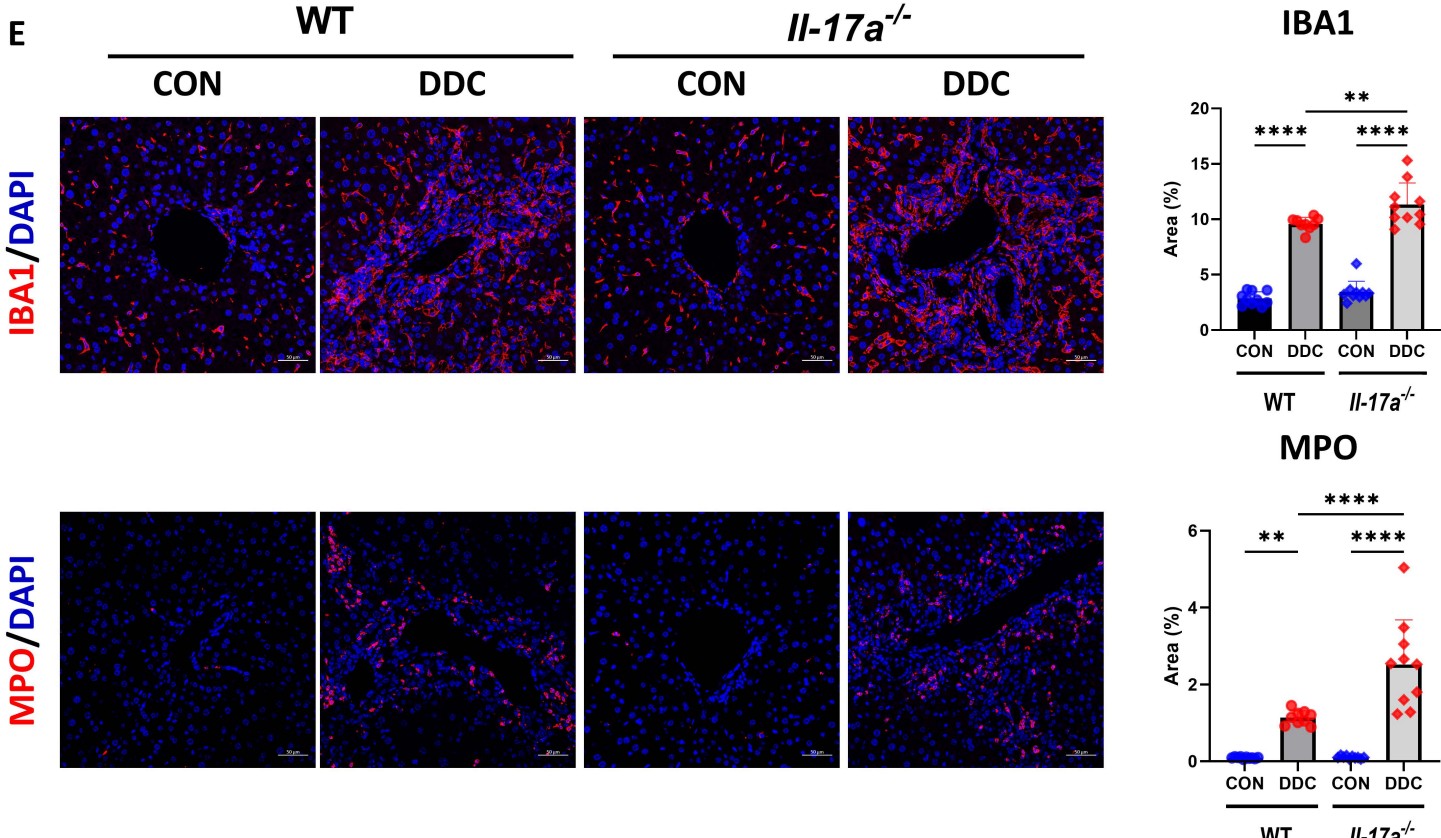

**Fig 4. DDC diet-induced cholestasis increases inflammation and drives a change in the hepatic immune phenotype. A**. t-SNE plot showing the spatial distribution of the 28 immune cell clusters resolved by unsupervised non-linear dimension reduction. Similar cells cluster together, while dissimilar cells are further apart. Cluster identity is enumerated below the t-SNE plot. **B. Left panel:** Hierarchical clustering heatmap of immune cell cluster abundance for all experimental animals is shown. WT and *Il-17a⁻/⁻* cholestatic mice clustered together while control diet fed mice clustered together illustrating that injury alters the immune landscape. **Right panel:** Hierarchical clustering heatmap of markers provides the immune cell signature of individual clusters. **C.** t-SNE plots of immune cell cluster abundance of each experimental group are presented (N = 3 per group). **D.** The abundance of predominant immune cell types (B cells, T cells, Kupffer cells, macrophages, neutrophils, and dendritic cells) is summed and graphically presented. **E. Upper panel:** Representative images of FPPE liver tissue sections stained for the pan-macrophage marker IBA1 (N = 10−11 per group). **Lower panel:** Representative images of FPPE liver tissue sections stained for the neutrophil marker, MPO (N = 6−9 per group). Digital image quantification for IBA-1 and MPO are provided in the corresponding panels on the right. Scale bar = 50 µm. * - p < 0.05, ** - p < 0.01, *** - p < 0.005, **** - p < 0.001.

Finally, primary human stellate cells stimulated with recombinant LIGHT induced the expression of profibrotic genes *ACTA2*, *COL1A1*, *PDGFRA and CTGF* (Fig 6J), suggesting that the action of LIGHT in fibroblast activation and fibrosis may be direct. These observations suggest an association between LIGHT expressing Th1 CD4⁺ T Cells and cholestatic liver fibrogenesis.

## Discussion

The results of this study provide key mechanical insights into the role of IL-17A modulation of cholestatic liver injury as assessed in the DDC murine model. The principal findings of this study are as follows: (1) The genetic deletion of *Il-17a* augments hepatic fibrosis; (2) NanoString analysis of the IHL population identified increased expression of the profibrogenic cytokine LIGHT; (3) CD4⁺ T cells were the primary source of LIGHT within the IHL population, with enhanced LIGHT identification in the Th1 CD4⁺ T cells. These observations are discussed in detail.

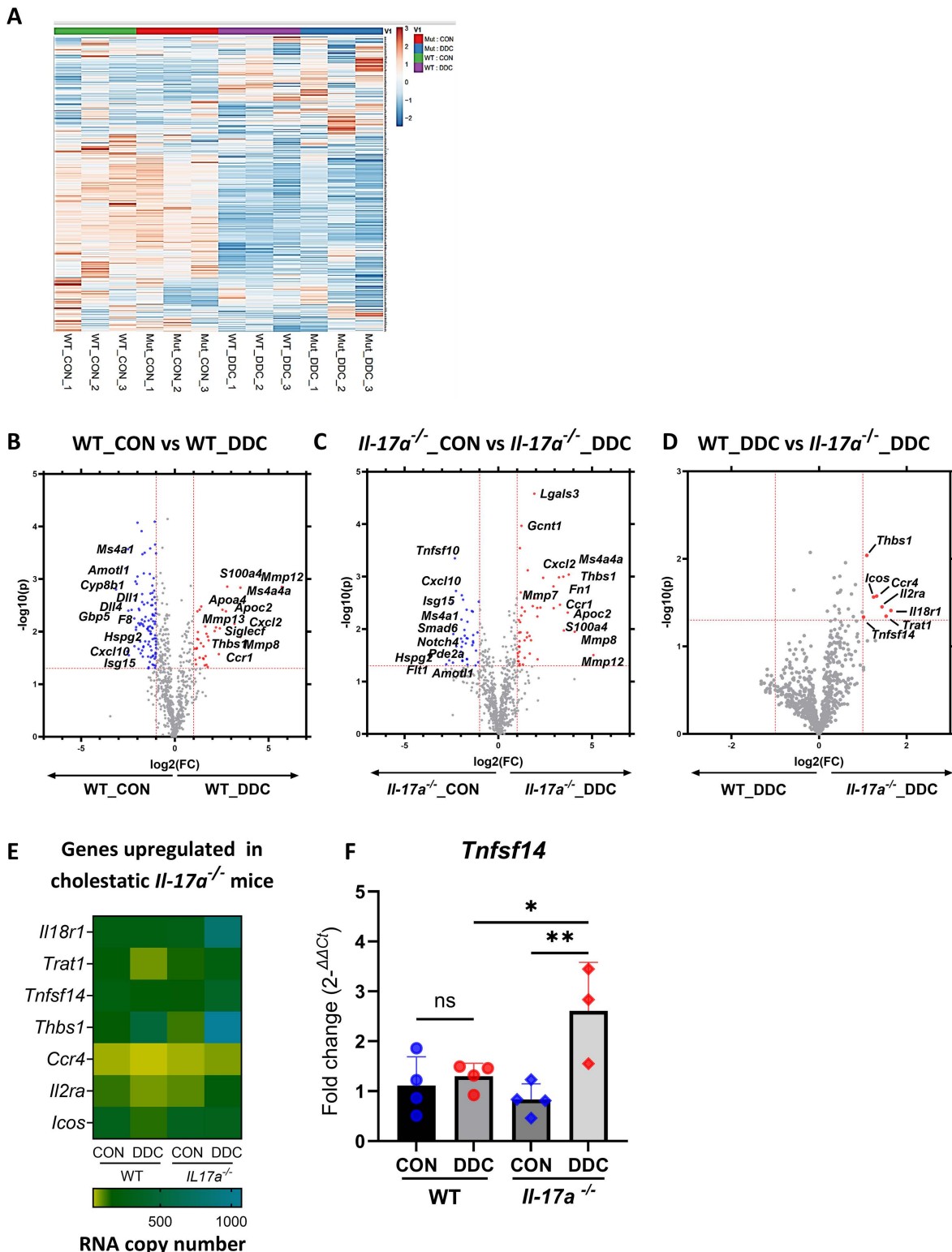

**Fig 5. Immune cells from cholestatic mice _Il-17a_-/- express a profibrogenic gene signature that includes LIGHT (_Tnfsf14_).** (**A**) Heatmap generated using Clustvis R showing the expression of 760 fibrosis related genes across the WT and mutant cholestatic mice (N = 3 per group). Volcano plots showing fold change in gene expression between (**B**) WT mice on control and DDC diet, (**C**) _Il-17a_-/- mice on control and DDC diets, and (**D**) WT and

*Il-17a⁻ᐟ⁻* mice on DDC diet. Genes significantly upregulated with cholestasis are shown in red, and genes significantly downregulated are in blue. (**E**) Heatmap constructed from the RNA copy number of the seven genes that were upregulated in cholestatic *Il-17a⁻ᐟ⁻* mice as compared to their WT counterparts. (**F**) Gene expression of *Tnfsf14* that encodes for the protein LIGHT in IHL is graphed (N = 3-4 per group). * - p < 0.05, ** - p < 0.01.

Murine models of cholestatic liver injury do not recapitulate all the features of human cholestatic liver disease. We elected to use the DDC murine model of cholestatic liver injury as this model elicits robust bile duct proliferation and periductal fibrosis prominent features of human cholestatic liver disease [28] Given our interest in the ductular reaction and periductal fibrosis, the DDC model is an appropriate model to assess the role of IL-17A in cholestatic fibrogenesis and fibrosis. However, a limitation of these studies is that our observations has not been established in additional models.

The germline deletion of *Il-17a* accentuated hepatic fibrosis without altering other features of cholestatic liver injury in DDC-fed mice. For example, serum biochemical parameters of cholestasis (e.g., serum ALP, bile acid, and TBIL values) were similar between WT and *Il-17a⁻ᐟ⁻* mice. This observation suggests that IL-17A restrains hepatic fibrogenesis directly or indirectly during cholestasis. Because IL-17A directly activates mouse hepatic stellate cells [17], its genetic deletion would be anticipated to reduce fibrosis, an effect contrary to our current observations. As IL-17A is an immune-modulatory cytokine, we profiled fibrogenic mediators in IHL and identified an increase in the profibrogenic cytokine, LIGHT, in the absence of IL-17A. The source of LIGHT was CD4⁺ T cells, as all subsets examined, including Th1- and Th2-polarized CD4⁺, Th17, and T regulatory cells expressed LIGHT. Hence, the effect of IL-17A in restraining fibrosis appears to be indirect by modulating CD4⁺ T cell effector function.

The increase in LIGHT expression during cholestasis was only observed in the Th1-polarized CD4⁺ T cells. This observation is consistent with observations reporting that the transcription factor NFAT promotes a Th1-like pattern of gene expression in primary murine CD4 T cells [27], and that NFAT is known to enhance LIGHT expression [29]. Moreover, LIGHT is known to induce TGFβ in macrophages which is pro-fibrogenic [30], and our studies demonstrate that LIGHT may directly activate primary HSC. Taken together, these data suggest that Th1-polarized CD4⁺ T cells may promote cholestatic fibrogenesis by expressing LIGHT. Whether LIGHT is sufficient to promote cholestatic fibrogenesis or whether it synergizes with other mediators as has been reported in pulmonary fibroblasts will require further delineation [31]. We also note that these studies only identify a relationship between LIGHT expression and cholestatic fibrogenesis and conditional LIGHT knockout mice, which have yet to be developed, would be necessary to further a mechanistic relationship.

Although well established to be profibrogenic in multiple tissues, LIGHT has not been well studied in liver injury or fibrosis. One study suggested LIGHT genetic deficiency ameliorated hepatic steatosis in mice fed a high-fat, high-cholesterol diet [32]. In contrast, another study correlated decreases in circulating LIGHT concentrations following splenectomy with hepatic fibrosis regression [33]. Recently, LIGHT was shown to drive a pro-inflammatory transcriptomic signature in pulmonary fibroblasts [31]. These data are consistent with the more ubiquitous expression of the LIGHT receptor on cell types responsible for tissue fibrosis [30,34]. We also note that LIGHT may be expressed by activated cholangiocytes in cholestasis [35,36] as another cellular source of this profibrogenic ligand. Nevertheless, in our study we confirmed that LIGHT can promote activation of pro-fibrotic gene expression in primary human hepatic stellate cells. The relative contribution of LIGHT as compared to other fibrogenic mediators, as well as the dominant cellular sources of this cytokine in cholestatic fibrogenesis, will require further exploration.

This study has three principal implications. First, anti-IL17 therapy targeting IL-17A and other family members in cholestatic liver disease such as PSC may not be advisable, given the results of this study demonstrating enhanced cholestatic fibrogenesis. Second, targeting CD4⁺ T cells to reduce hepatic fibrosis in cholestatic liver diseases besides PSC requires further exploration. Finally, the contribution of LIGHT to hepatic fibrosis appears to be important; hence, functionally inhibiting the action of this cytokine is a potential strategy for reducing hepatic fibrosis.

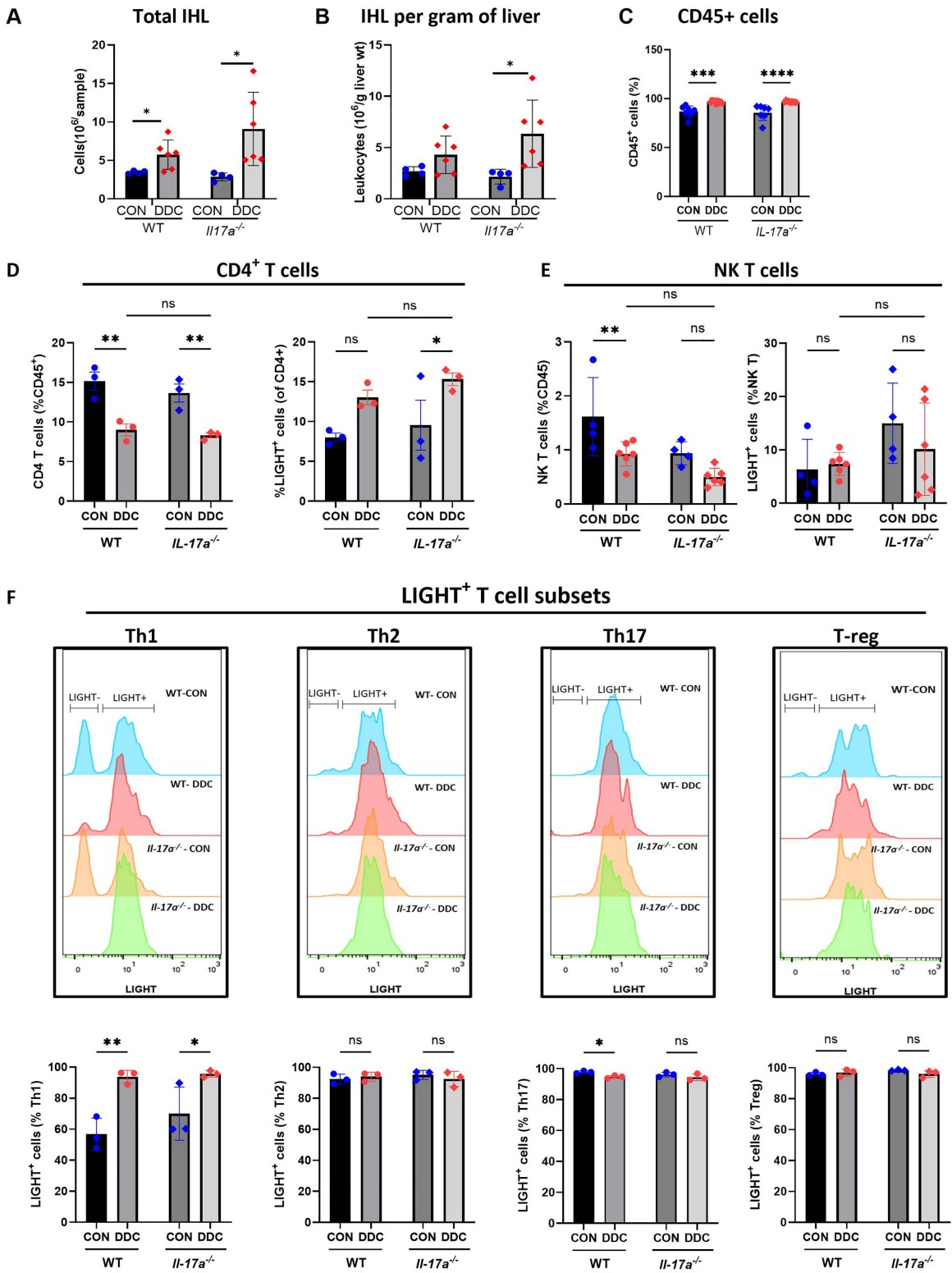

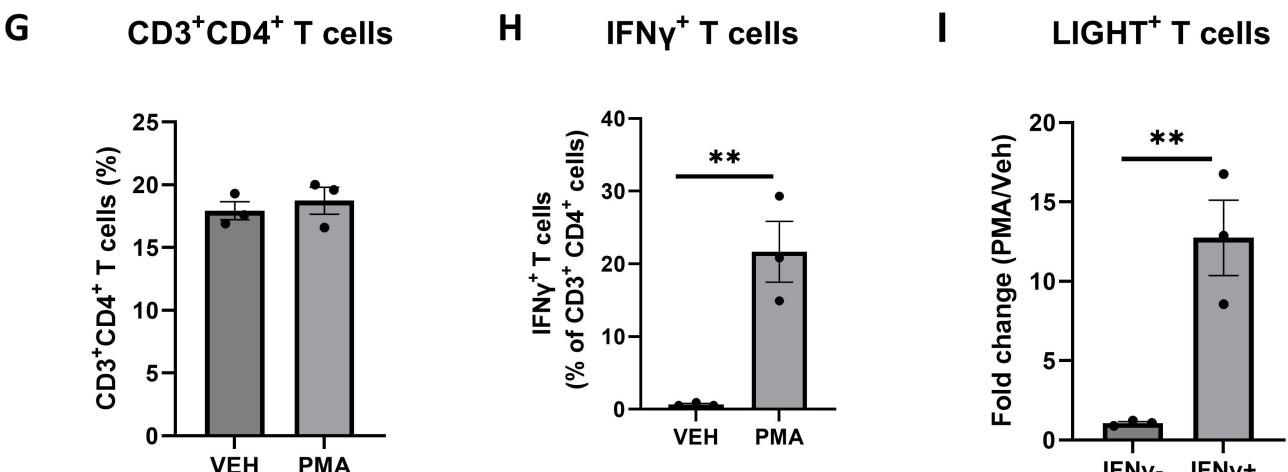

## J Expresssion of pro-fibrotic genes in primary human fibroblasts

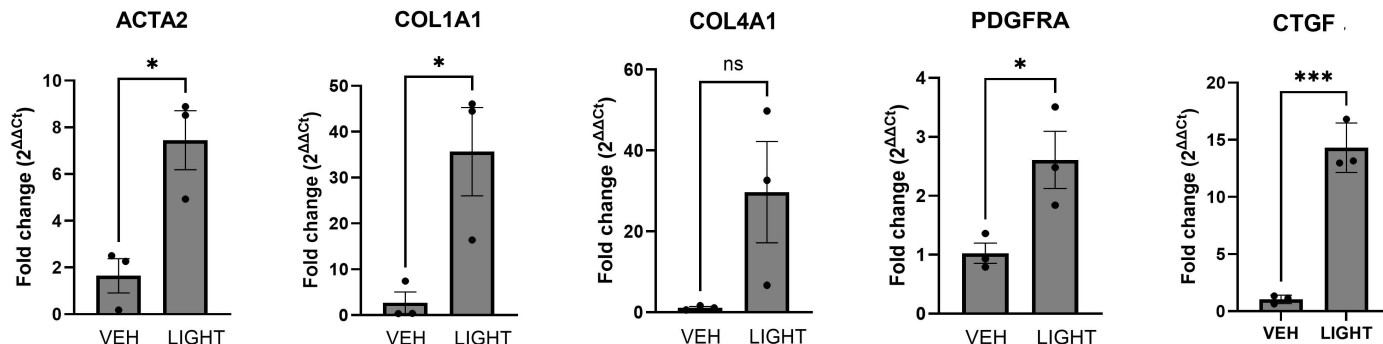

**Fig 6. CD4+ T cells express LIGHT. A.** The abundance of total IHL isolated from WT and *Il-17a-/-* mice is quantified (N=4-6 per group). **B.** Total number of IHL isolated from WT and *Il-17a-/-* mice reared on control or DDC diet is normalized to liver weight (N=4-6 per group). **C.** The proportion of CD45+ cells in the IHL as determined by flow cytometry (N=4-6 per group). **D. (Left panel)** CD4+ T cells abundance in gated CD45+ immune cells, and **(Right panel)** the proportion of LIGHT+ cells in the CD4+ T cell fraction (N=3 per group). **E. (Left panel)** NK T cell abundance in gated CD45+ immune cells, and **(Right panel)** proportion of LIGHT+ cells in the NK T cell fraction (N=4-6 per group). **F.** LIGHT positivity in CD4+ T cell subsets, Th1, Th2, Th17, and T-reg, was determined by flow cytometry (N=3 per group). **G. WT** primary mouse splenocytes when stimulated with PMA/ionomycin or vehicle display (left panel) a similar proportion of CD3+CD4+ T cells, (middle panel) an increased abundance of IFNγ+ CD 4+ T cells and (right panel) an increased proportion of LIGHT+ CD4+ T cells (N=3 per group). **H.** The expression of pro-fibrogenic genes *ACTA2, COL1A1, PDGFRA* and *CTGF* is elevated in primary human hepatic stellate cells when stimulated with recombinant LIGHT (N=3 per group). * - p<0.05, ** - p<0.01, *** - p<0.005, **** - p<0.001.

## Supporting information

**S1 File. Data set.**

(XLSX)

**S1 Fig. Genotyping of Il-17a-/- mice.**

(TIF)

**S2 Fig. Immune cells and fibroblasts in the periportal zone of a DDC diet-induced cholestatic WT mouse.**
(TIF)

**S3 Fig. Cell cluster identification of intrahepatic leukocytes using mass cytometry.**
(TIF)

**S4 Fig. Percentage of immune cell clusters in total IHL as determined by mass cytometry.**
(TIF)

**S5 Fig. A. Detection of LIGHT$^+$ immune cells by flow cytometry. B-F. Detection of LIGHT$^+$ immune cells by flow cytometry.**
(ZIP)

**S6 Fig. LIGHT$^+$ cells in the CD3$^+$CD4$^+$IFNγ$^+$ subpopulation of WT and *Il17a*-/- mice splenocytes.**
(TIF)

**S1 Table. List of antibodies used for immunofluorescent staining of tissue sections and for flow cytometry.**
(DOCX)

**S2 Table. List of primers used for quantitative real-time PCR.**
(DOCX)

**S3 Table. List of antibodies and clones used for mass cytometry.**
(DOCX)

## Acknowledgments

The administrative assistance of Ms. Courtney Hoover is greatly appreciated.

## Author contributions

**Conceptualization:** Anuradha Krishnan, Maria Eugenia Guicciardi, Adiba I. Azad, Gregory J Gores.

**Data curation:** Takashi Kitagataya, Anuradha Krishnan, Kirsta E. Olson, Florencia Gutierrez, Michelle Baez-Faria, Maria Eugenia Guicciardi, Adiba I. Azad.

**Formal analysis:** Takashi Kitagataya, Kirsta E. Olson, Florencia Gutierrez, Maria Eugenia Guicciardi, Kevin D. Pavelko, Adiba I. Azad, Gregory J Gores.

**Funding acquisition:** Gregory J Gores.

**Investigation:** Anuradha Krishnan, Maria Eugenia Guicciardi, Kevin D. Pavelko, Adiba I. Azad, Gregory J Gores.

**Methodology:** Kevin D. Pavelko, Adiba I. Azad, Gregory J Gores.

**Project administration:** Kirsta E. Olson, Gregory J Gores.

**Resources:** Gregory J Gores.

**Writing – original draft:** Takashi Kitagataya, Anuradha Krishnan, Kirsta E. Olson, Florencia Gutierrez, Michelle Baez-Faria, Maria Eugenia Guicciardi, Kevin D. Pavelko, Adiba I. Azad, Gregory J Gores.

**Writing – review & editing:** Takashi Kitagataya, Anuradha Krishnan, Kirsta E. Olson, Florencia Gutierrez, Michelle Baez-Faria, Maria Eugenia Guicciardi, Kevin D. Pavelko, Adiba I. Azad, Gregory J Gores.

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
