## [Decision Letter · Decision Letter 0]

20 Oct 2025

Dear Dr. Gores,

Thank you for submitting your manuscript to PLOS ONE. After careful consideration, we feel that it has merit but does not fully meet PLOS ONE’s publication criteria as it currently stands. Therefore, we invite you to submit a revised version of the manuscript that addresses the points raised during the review process.

We look forward to receiving your revised manuscript.

Kind regards,

Pavel Strnad

Academic Editor

PLOS ONE

Journal Requirements:

“This work was supported by NIDDK-funded grant DK124182 (GJG), the NIDDK-funded Optical Microscopy Core of the Mayo Clinic Center for Cell Signaling in Gastroenterology (P30DK084567), and the Mayo Clinic, Rochester.”

3. Please expand the acronym “NIDDK” (as indicated in your financial disclosure) so that it states the name of your funders in full.

“Disclosures: The authors have no financial or personal disclosures relevant to this manuscript.”

5. We note that your Data Availability Statement is currently as follows: Data supporting the findings of this study are available within the article and in the supplementary information.

Additional Editor Comments:

As you can see, both reviewers appreciated your work, I am looking forward to the revised version!

Reviewers' comments:

Reviewer's Responses to Questions

**Comments to the Author**

1. Is the manuscript technically sound, and do the data support the conclusions?

Reviewer #1: No

Reviewer #2: Yes

2. Has the statistical analysis been performed appropriately and rigorously?

Reviewer #1: No

Reviewer #2: Yes

3. Have the authors made all data underlying the findings in their manuscript fully available?

Reviewer #1: Yes

Reviewer #2: Yes

4. Is the manuscript presented in an intelligible fashion and written in standard English?

Reviewer #1: Yes

Reviewer #2: Yes

Reviewer #1: This study attempts to dissect the role of IL-17A in cholestatic liver injury using a DDC-induced mouse model and proposes an intriguing counterintuitive role for IL-17A as anti-fibrogenic, via suppression of the LIGHT(TNFSF14) axis in Th1 CD4+ T cells. The hypothesis is original, and the data volume is substantial. However, several critical flaws exist that undermine the rigor, clarity, and significance of the work.

Major Criticisms

1. Used model possesses critical limitations that undermine translational relevance. The study exclusively uses the DDC model, which does not truly model human cholangiopathies like PSC or PBC. It primarily induces bile duct proliferation and periductal fibrosis, but lacks the chronicity and immune features (e.g., autoantibodies, IBD comorbidity) found in PSC. No validation in a second model (e.g., BDL, Mdr2-/-) is a serious oversight, especially since the authors make clinical inferences about anti-IL-17 therapy in PSC. The discussion acknowledges this limitation, but this should not just be noted — it weakens the core claims of the manuscript.

2. In fact, LIGHT mechanism is not directly demonstrated in vivo. LIGHT expression is shown to be upregulated in CD4+ Th1 cells in flow cytometry, and LIGHT can activate HSCs in vitro — but no causal link is shown in vivo. There is no use of LIGHT-deficient mice, blocking antibodies, or neutralization strategies. No functional experiment demonstrates that the fibrosis seen in Il-17a-/- mice is actually due to LIGHT — this is speculative at best. A correlative upregulation is not a mechanistic demonstration.

3. IL-17A is known to have complex, context-dependent roles in mucosal immunity and fibrosis. The authors cite this but still oversimplify by declaring IL-17A as "restraining" fibrosis via LIGHT regulation. Th1 skewing is assumed based on increased LIGHT in Th1 cells, but no cytokine profile (e.g., IFNγ, IL-2) is provided to support Th1 dominance. Other CD4+ subsets (e.g., Th17, Treg) are shown to express LIGHT, but the authors arbitrarily focus on Th1 cells.

4. Although CyTOF is used to generate 28 clusters, the analysis is descriptive and shallow: no statistical clustering (e.g., PCA, UMAP, cluster-specific testing), no deep phenotype mapping beyond broad lineage categories, no correlation between specific clusters and fibrosis severity.

5. Many key figures (especially immunostaining) are poorly quantified or not quantified at all. No available description of the sample number per experiment per quantification. No scoring system or blinded analysis is mentioned for fibrosis, desmin, CK19. Quantification of Sirius Red is questionable — was this done per portal area, per field, or whole slide? Quantitative rigor is lacking throughout the manuscript.

6. Key Controls are missing such as: No IL-17F single KO or IL-17A/F double KO controls used in key experiments. No isotype controls shown in flow cytometry for LIGHT or transcription factor staining. No cell-specific knockout models used to prove LIGHT source or action. No cytokine stimulation of CD4+ T cells to show IL-17A suppresses LIGHT.

Reviewer #2: The study by Kitagataya et al., titled “Genetic Ablation of IL17A Augments Fibrosis in a Mouse Model of Cholestatic Liver Injury,” is highly comprehensive and methodologically robust. The concept that this pathway influences fibrosis by inhibiting LIGHT and might have potential for future therapeutic applications.

From structural perspective, the manuscript is fluent, and it is clear that the authors have provided key details regarding materials and methods. However, several aspects could be improved for better clarity and scientific rigor.

The experimental design is generally strong, with well-connected experiments that illuminate the underlying mechanisms. Nonetheless, specific key findings should be further investigated and/or discussed.

My specific comments are provided below:

1- In some analyses, precise statistical evaluations are missing In Figure 1D–E, statistical comparisons are shown for WT vs WT-DDCfed and Il17A KO vs IL17A-DDCfed mice; however, it is unclear whether there is also a statistical difference between WT-fed and KO-fed mice. Additionally, in Figure 1B, the statistical significance is not clearly indicated.

2- Most of the figures are difficult to read due to small font sizes and low image resolution, making it challenging to interpret the data. There are general resolution problems in the microscopic images. For instance, in Figure 1B, CK19 staining is barely visible, and the scale bars are inconsistent. A similar issue is observed in Figure 2C and several other figures. In some figures (e.g., Figure 5A–B, Figures S2–S3), the legends and labels are also hard to read.

3- Previous studies have shown that LIGHT enhances TGF-β production and fibrotic gene expression via the LTβR–NF-κB pathway. However, this study does not examine whether LIGHT exerts its effect through this signaling route. similarly, it remains unclear whether the STAT1 pathway was investigated, although it could be mechanistically relevant.

4- Furthermore, the observation that LIGHT elevation shifts polarization toward the T1 phenotype raises questions about which cytokines may mediate this process. For example, were IFN-γ or other related cytokines analyzed?

5- The discussion section would benefit from a more integrative approach one that connects these possible mechanisms and situates the findings within the broader context of LIGHT-related studies. Specifically, the authors should discuss how T-cell polarization in the liver may be influenced by LIGHT, and how the NF-κB and STAT pathways could be involved in these interactions.

**Do you want your identity to be public for this peer review?** For information about this choice, including consent withdrawal, please see our Privacy Policy

Reviewer #1: No

Reviewer #2: No

---

## [Author Response · Author response to Decision Letter 1]

2 Jan 2026

please see attached, detailed response to reviewers document

---

## [Decision Letter · Decision Letter 1]

9 Jan 2026

Dear Dr. Gores,

We look forward to receiving your revised manuscript.

Kind regards,

Pavel Strnad

Academic Editor

PLOS One

Journal Requirements:

Reviewers' comments:

Reviewer's Responses to Questions

**Comments to the Author**

Reviewer #1: All comments have been addressed

Reviewer #2: (No Response)

2. Is the manuscript technically sound, and do the data support the conclusions?

Reviewer #1: Yes

Reviewer #2: Partly

3. Has the statistical analysis been performed appropriately and rigorously?

Reviewer #1: Yes

Reviewer #2: Yes

4. Have the authors made all data underlying the findings in their manuscript fully available?

Reviewer #1: Yes

Reviewer #2: Yes

5. Is the manuscript presented in an intelligible fashion and written in standard English?

Reviewer #1: Yes

Reviewer #2: Yes

Reviewer #1: The authors have largely addressed the comments raised during peer review. The revised version clarifies key aspects of the proposed mechanism and more appropriately frames the findings within the limitations of the experimental approach. The conclusions are generally consistent with the data presented.

However, several additional points could further strengthen the manuscript, although are not critical for acceptance. First, the absence of in vivo loss-of-function experiments targeting LIGHT (e.g., antibody-mediated blockade) is understandable given practical limitations; while such experiments would provide stronger mechanistic support, their omission does not fundamentally compromise the main conclusions, which are presented in a suitably cautious manner.

Second, IL-17F is evaluated only at the transcript level. As mRNA expression does not necessarily reflect protein abundance and IL-17F could theoretically contribute to compensatory signaling, this limitation should be acknowledged more explicitly. Nevertheless, it does not materially alter the interpretation of the findings.

In addition, it would be important to clearly state as conclusion that the reported effects are specific to the DDC-induced cholestasis model and should not be generalized to other forms of liver injury or fibrosis without further validation.

Finally, for clarity and transparency, it would be appropriate for the authors to explicitly state the number of biological replicates (n) in each figure legend.

Overall, I find the revision acceptable, and the points above should be considered minor recommendations rather than prerequisites for acceptance.

Reviewer #2: The authors have addressed the majority of the reviewers’ comments, and the study appears scientifically sound. The presented data support the main conclusions. Overall, I consider the work suitable for acceptance after minor revisions.

Nevertheless, a few points could further strengthen the manuscript:

1. The authors discuss their observations on LIGHT⁺ Th1 T cells in the context of DDC diet–induced cholestatic liver injury and fibrosis. However, the mechanistic aspects underlying these observations remain relatively limited. A more detailed discussion of the potential cellular and molecular mechanisms by which LIGHT⁺ Th1 T cells may contribute to fibrosis would substantially strengthen the Discussion.

2. Gene names are inconsistently formatted throughout the manuscript (e.g., COL1a1 on page 10). Please ensure consistent and correct gene nomenclature according to standard conventions.

3. The resolution of some figures remains suboptimal. In particular, Figure 5b would benefit from higher resolution to improve readability. In Figure 3, the immunofluorescence signals appear weak and are difficult to distinguish. Improving image contrast and resolution, or providing higher-magnification images, would help readers better appreciate the findings.

**Do you want your identity to be public for this peer review?** For information about this choice, including consent withdrawal, please see our Privacy Policy

Reviewer #1: No

Reviewer #2: No

---

## [Author Response · Author response to Decision Letter 2]

19 Jan 2026

RESPONSES TO COMMENTS BY REVIEWER NUMBER 1

We thank the reviewer for his/her thoughtful and constructive examination of our manuscript. We have addressed all the reviewer’s concerns in detail. Our responses to the reviewer’s comments are as follows:

COMMENT 1. Several additional points could further strengthen the manuscript, although are not critical for acceptance. First, the absence of in vivo loss-of-function experiments targeting LIGHT (e.g., antibody-mediated blockade) is understandable given practical limitations; while such experiments would provide stronger mechanistic support, their omission does not fundamentally compromise the main conclusions, which are presented in a suitably cautious manner.

RESPONSE: We thank the reviewer’s acknowledgement of the experimental limitations of the above-mentioned experiment and for appreciating that the results are presented with caution.

COMMENT 2. IL-17F is evaluated only at the transcript level. As mRNA expression does not necessarily reflect protein abundance and IL-17F could theoretically contribute to compensatory signaling, this limitation should be acknowledged more explicitly. Nevertheless, it does not materially alter the interpretation of the findings.

RESPONSE: We acknowledge that IL-17F was determined only at the transcript level. This has been addressed explicitly on lines 234-235 of the manuscript as follows: “The absence of compensation by Il-17f was also excluded by qRT-PCR for its mRNA, albeit this was not confirmed at the protein level (Fig. 1C).”

COMMENT 3. In addition, it would be important to clearly state as conclusion that the reported effects are specific to the DDC-induced cholestasis model and should not be generalized to other forms of liver injury or fibrosis without further validation.

RESPONSE: The conclusion of the abstract (lines 52-54) has been revised to state “Taken together, these data suggest that IL-17A restrains expression of the profibrogenic cytokine, LIGHT, by Th1-polarized CD4+ T cells, and implicate a role for LIGHT in cholestatic fibrogenesis in DDC-fed mice; a finding which requires validation in additional models.”

COMMENT 4. Finally, for clarity and transparency, it would be appropriate for the authors to explicitly state the number of biological replicates (n) in each figure legend.

RESPONSE: The number of biological replicates relevant to individual panels within each figure has been included in the legend.

RESPONSES TO COMMENTS BY REVIEWER NUMBER 2

We thank the reviewer for his/her thoughtful and constructive examination of our manuscript. We have addressed all the reviewer’s concerns in detail. Our responses to the reviewer’s comments are as follows:

COMMENT 1. The authors discuss their observations on LIGHT⁺ Th1 T cells in the context of DDC diet–induced cholestatic liver injury and fibrosis. However, the mechanistic aspects underlying these observations remain relatively limited. A more detailed discussion of the potential cellular and molecular mechanisms by which LIGHT⁺ Th1 T cells may contribute to fibrosis would substantially strengthen the Discussion.

RESPONSE: The discussion has been updated to include potential mechanistic aspects of the contribution of LIGHT to fibrosis. The discussion on lines 435-442 reads as follows: “This observation is consistent with observations reporting that the transcription factor NFAT promotes a Th1-like pattern of gene expression in primary murine CD4 T cells (PMID11994444), and that NFAT is known to enhance LIGHT expression (PMID 12215452). Moreover, LIGHT is known to induce TGFβ in macrophages which is pro-fibrogenic (PMID 37465675), and our studies demonstrate that LIGHT may directly activate primary HSC. Taken together, these data suggest that Th1-polarized CD4+ T cells may promote cholestatic fibrogenesis by expressing LIGHT. Whether LIGHT is sufficient to promote cholestatic fibrogenesis or whether it synergizes with other mediators as has been reported in pulmonary fibroblasts will require further delineation (PMID 40972652)”.

COMMENT 2. Gene names are inconsistently formatted throughout the manuscript (e.g., COL1a1 on page 10). Please ensure consistent and correct gene nomenclature according to standard conventions.

RESPONSE: We thank the reviewer for pointing this out. We have reviewed the manuscript and ensured that the gene nomenclature is per standard convention.

COMMENT 3. The resolution of some figures remains suboptimal. In particular, Figure 5b would benefit from higher resolution to improve readability. In Figure 3, the immunofluorescence signals appear weak and are difficult to distinguish. Improving image contrast and resolution, or providing higher-magnification images, would help readers better appreciate the findings.

RESPONSE: In Figure 5B, the figure size and font size have both been enlarged to improve the readability of individual gene names within the panel. In Figure 3, immunofluorescence in the red channel has been pseudo-colored to yellow to enhance contrast and improve visibility.

---

## [Editor Report · Decision Letter 2]

20 Jan 2026

Genetic Ablation of Interleukin 17A Augments Fibrosis in a Mouse Model of Cholestatic Liver Injury

PONE-D-25-52210R2

Dear Dr. Gores,

We’re pleased to inform you that your manuscript has been judged scientifically suitable for publication and will be formally accepted for publication once it meets all outstanding technical requirements.

Kind regards,

Pavel Strnad

Academic Editor

PLOS One

Additional Editor Comments (optional): Thank you for your great work!
---

## [Editor Report · Acceptance letter]

PONE-D-25-52210R2

PLOS One

Dear Dr. Gores,

I'm pleased to inform you that your manuscript has been deemed suitable for publication in PLOS One. Congratulations! Your manuscript is now being handed over to our production team.

Kind regards,

on behalf of

Dr. Pavel Strnad

Academic Editor

PLOS One